

# *Bax-inhibitor-1* knockdown phenotypes are suppressed by *Buffy* and exacerbate degeneration in a *Drosophila* model of Parkinson disease

P. Githure M'Angale and Brian E. Staveley

Department of Biology, Memorial University of Newfoundland, St. John's, NL, Canada

## ABSTRACT

**Background:** Bax inhibitor-1 (BI-1) is an evolutionarily conserved cytoprotective transmembrane protein that acts as a suppressor of *Bax*-induced apoptosis by regulation of endoplasmic reticulum stress-induced cell death. We knocked down *BI-1* in the sensitive *dopa decarboxylase* (*Ddc*) expressing neurons of *Drosophila melanogaster* to investigate its neuroprotective functions. We additionally sought to rescue the *BI-1*-induced phenotypes by co-expression with the pro-survival *Buffy* and determined the effect of *BI-1* knockdown on the neurodegenerative α-*synuclein*-induced Parkinson disease (PD) model.

**Methods:** We used organismal assays to assess longevity of the flies to determine the effect of the altered expression of *BI-1* in the *Ddc-Gal4*-expressing neurons by employing two RNAi transgenic fly lines. We measured the locomotor ability of these RNAi lines by computing the climbing indices of the climbing ability and compared them to a control line that expresses the *lacZ* transgene. Finally, we performed biometric analysis of the developing eye, where we counted the number of ommatidia and calculated the area of ommatidial disruption.

**Results:** The knockdown of *BI-1* in these neurons was achieved under the direction of the *Ddc-Gal4* transgene and resulted in shortened lifespan and precocious loss of locomotor ability. The co-expression of *Buffy*, the Drosophila anti-apoptotic Bcl-2 homologue, with *BI-1-RNAi* resulted in suppression of the reduced lifespan and impaired climbing ability. Expression of human α-*synuclein* in Drosophila dopaminergic neurons results in neuronal degeneration, accompanied by the age-dependent loss in climbing ability. We exploited this neurotoxic system to investigate possible BI-1 neuroprotective function. The co-expression of α-*synuclein* with *BI-1-RNAi* results in a slight decrease in lifespan coupled with an impairment in climbing ability. In supportive experiments, we employed the neuron-rich Drosophila compound eye to investigate subtle phenotypes that result from altered gene expression. The knockdown of *BI-1* in the Drosophila developing eye under the direction of the *GMR-Gal4* transgene results in reduced ommatidia number and increased disruption of the ommatidial array. Similarly, the co-expression of *BI-1-RNAi* with *Buffy* results in the suppression of the eye phenotypes. The expression of α-*synuclein* along with the knockdown of *BI-1* resulted in reduction of ommatidia number and more disruption of the ommatidial array.

**Conclusion:** Knockdown of *BI-1* in the dopaminergic neurons of Drosophila results in a shortened lifespan and premature loss in climbing ability, phenotypes that

Corresponding author
Brian E. Staveley, bestave@mun.ca

appear to be strongly associated with models of PD in Drosophila, and which are suppressed upon overexpression of *Buffy* and worsened by co-expression with α-*synuclein*. This suggests that *BI-1* is neuroprotective and its knockdown can be counteracted by the overexpression of the pro-survival *Bcl-2* homologue.

## INTRODUCTION

Bax inhibitor-1 (BI-1) belongs to a diverse group of proteins, known as transmembrane Bax inhibitor-1 motif-containing (TMBIM) family (*Henke et al., 2011*; *Li et al., 2014*; *Reimers et al., 2008*; *Rojas-Rivera & Hetz, 2015*), that have been determined to be regulators of cell death. A different nomenclature categorises these proteins into the *LFG* family, adopted from the family member *Lifeguard* (*Hu, Smith & Goldberger, 2009*), which consists of at least six highly conserved members in a wide range of organisms (*Chae et al., 2003*; *Henke et al., 2011*; *Huckelhoven, 2004*). These regulators of cell death, accomplish this role by the regulation of the death receptor, modulation of the endoplasmic reticulum (ER) calcium homeostasis, ER stress signalling pathways, autophagy, reactive oxygen species (ROS) production, cytosolic acidification, and other cellular activities (*Li et al., 2014*; *Rojas-Rivera & Hetz, 2015*). The founding member of this group is *BI-1*, or *TMBIM6* also known as *testis enhanced gene transcript*, and has been demonstrated to inhibit the effect of *Bax*-induced cell death (*Walter et al., 1995*; *Xu & Reed, 1998*). Members of this protein family possess a BI-1-like domain with six to seven transmembrane (TM)-spanning regions that are strongly associated with the ER membranes (*Carrara et al., 2012*; *Chae et al., 2004*; *Xu & Reed, 1998*). *BI-1* is highly conserved across diverse species with eukaryotic homologues of *BI-1* able to block *Bax*-induced cell death when expressed in yeast (*Chae et al., 2003*), thus implying it regulates an evolutionarily conserved cytoprotective pathway.

This protein though not structurally related to the *B cell lymphoma 2* family of proteins, forms a complex with the pro-survival members Bcl-2 and Bcl-X$_L$ but not with Bax or Bak (*Lisbona et al., 2009*; *Xu & Reed, 1998*). Therefore, it is likely the anti-apoptotic activity of *BI-1/TMBIM6* is mediated by interaction with pro-survival members of the *Bcl-2* family and acts downstream of Bcl-X$_L$ (*Xu et al., 2008*). *BI-1* deficient cells, that include neurons, are more sensitive to apoptosis induced by ER stress and has been linked to the modulation of ER calcium homeostasis (*Chae et al., 2004*; *Dohm et al., 2006*). This implicates BI-1 in a variety of human diseases that include numerous cancers, obesity, liver diseases, autoimmune response, and diabetes (*Kiviluoto et al., 2012*; *Li et al., 2014*; *Lisak et al., 2016*; *Robinson et al., 2011*; *Rojas-Rivera & Hetz, 2015*). Neuroprotective roles include, protection from oxygen–glucose deprivation, promotion of neuronal proliferation and differentiation, and stress-induced protection (*Dohm et al., 2006*; *Hunsberger et al., 2011*; *Jeon et al., 2012*; *Krajewska et al., 2011*). It regulates ROS production by modulation of unfolded protein response (UPR)

induction in the ER (*Lee et al., 2007*), suppression of mitochondria-mediated ROS production (*Kim et al., 2012*), reduction of cytochrome P450 2E1 activity, and regulation of the ER membrane lipid peroxidation (*Kim et al., 2009*). BI-1 undoubtedly has significant cytoprotective roles and their abrogation lead to cellular homeostatic dysfunction and disease.

*Drosophila melanogaster* appear to possess most of the TMBIM protein family homologues with TMBIM6/BI-1 represented by *BI-1/CG7188* (*Attrill et al., 2015*; *Hu, Smith & Goldberger, 2009*; *Rojas-Rivera & Hetz, 2015*). Drosophila has been used as a model organism in the study of gene expression and in human disease models, albeit with very promising results (*Staveley, 2015*). Several studies have used Drosophila to elucidate the importance of this protein in cellular homeostasis; including functional conservation of this protein in evolutionarily diverse organisms (*Chae et al., 2003*), BI-1 as a negative regulator of the ER stress sensor IRE1α and its role in the UPR (*Lisbona et al., 2009*), and its modulation of autophagy (*Castillo et al., 2011*). Expression in the *Ddc-Gal4*-expressing neurons is the focus of our studies as they are very sensitive to subtle differences in gene products and can be used to study ROS, ER stress, apoptosis, autophagy, and many other cellular processes. This is mainly because they degenerate in an age-dependent manner and this degeneration manifests as deficiency in locomotor function (*Botella et al., 2009*; *Feany & Bender, 2000*; *Park, Schulz & Lee, 2007*; *Staveley, 2015*). The key elements of the Drosophila model of Parkinson disease (PD) that utilizes the expression of a human *α-synuclein* transgene to induce the PD-like symptoms (*Feany & Bender, 2000*); is its ability to recapitulate some features of human PD that include, age-dependent loss of dopaminergic (DA) neurons that manifest in age-dependent loss in locomotor function (*Auluck et al., 2002*; *Botella et al., 2009*; *Buttner et al., 2014*; *Feany & Bender, 2000*; *Kong et al., 2015*; *Staveley, 2015*; *Wang et al., 2015*; *Zhu et al., 2016*). The spatio-temporal *UAS/GAL4* expression system (*Brand & Perrimon, 1993*), and the availability of a plethora of promoters or enhancers of which *TH-Gal4*, *elav-Gal4*, and *Ddc-Gal4* are employed to model PD in flies.

The *Bcl-2* family member homologues in Drosophila are limited to the single anti-apoptotic *Buffy* and the pro-apoptotic *Debcl* (*Colussi et al., 2000*). In previous studies, the overexpression of *Buffy* has been shown to confer survival advantages specifically in response to external stimuli and in conditions of cellular stress (*M'Angale & Staveley, 2016a*; *Monserrate, Chen & Brachmann, 2012*; *Sevrioukov et al., 2007*; *Tanner et al., 2011*). This point to an important role for this protein in aspects of cell death. We investigated the outcome of the knockdown of *BI-1* in Drosophila neurons, and further determined whether there is an interaction with the anti-apoptotic Bcl-2 protein Buffy. We employed two different RNAi lines to determine the specificity of the effects of knockdown of this gene and compared them to a control line. We further co-expressed *BI-1* in DA neurons along with *α-synuclein* to investigate whether it possesses neuroprotective functions by assessing the phenotypes that would result from knockdown of *BI-1* and expression of *α-synuclein*. Lastly, in supportive experiments we attempted to establish a role for BI-1 in the Drosophila developing eye.

## MATERIALS AND METHODS

### Bioinformatic analysis

The protein sequences were obtained from the National Center for Biotechnology Information (NCBI; http://www.ncbi.nlm.nih.gov/protein/) and the domains were identified using the NCBI Conserved Domain Database (CDD; http://www.ncbi.nlm.nih.gov/cdd) (*Marchler-Bauer et al., 2015*) and the Eukaryotic Linear Motif (ELM; http://elm.eu.org/) (*Dinkel et al., 2016*) which focuses on annotation and detection of eukaryotic linear motifs (ELMs), also known as short linear motifs. A multiple sequence alignment was done using Clustal Omega (http://www.ebi.ac.uk/Tools/msa/clustalo/) (*Goujon et al., 2010*; *Sievers et al., 2011*) to show conservation of the domains in the selected organisms. The prediction of the nuclear export signal (NES) was by NetNES (http://www.cbs.dtu.dk/services/NetNES/) (*la Cour et al., 2004*). Further analysis of protein sequences was performed with Phyre2 (*Kelley et al., 2015*), a web portal for protein modelling, prediction, and analysis (http://www.sbg.bio.ic.ac.uk/phyre2/html/page.cgi?id=index). The sub-cellular localisation was performed by MultiLoc2 (*Blum, Briesemeister & Kohlbacher, 2009*) (https://abi.inf.uni-tuebingen.de/Services/MultiLoc2). Transmembrane domains were further investigated and identified using TMpred (*Artimo et al., 2012*), a program based on statistical analysis of TMbase (http://www.ch.embnet.org/software/TMPRED_form.html).

### Drosophila media, stocks, and derivative lines

Stocks and crosses were maintained on standard cornmeal/molasses/yeast/agar media treated with propionic acid and methylparaben to inhibit fungal growth. Stocks were kept at room temperature while crosses and experiments for analysis of ageing and climbing ability were carried out at 25 °C while those for the eye analysis were performed at 29 °C.

The P{KK100983}VIE-260B stock hereby referred to as UAS-BI-1-RNAi (1) (http://flybase.org/reports/FBst0481930.html) and w1118; P{GD1660}v37108 hereby referred to as UAS-BI-1-RNAi (2) (http://flybase.org/reports/FBst0461842.html) (*Dietzl et al., 2007*) were obtained from Vienna Drosophila Resource Center. Additional information on the RNAi constructs can be obtained from http://www.flyrnai.org/up-torr/. The *UAS-Buffy* (*Quinn et al., 2003*) was provided by Dr L. Quinn (University of Melbourne), *Ddc-Gal4* flies (*Li et al., 2000*) by Dr J. Hirsch (University of Virginia), and *UAS-α-synuclein* (*Feany & Bender, 2000*) by Dr M. Feany (Harvard Medical School). *GMR-Gal4* (*Freeman, 1996*) and *UAS-lacZ* flies were obtained from the Bloomington Drosophila Stock Center.

The *UAS-α-synuclein/CyO; Ddc-Gal4/TM3; UAS-α-synuclein/CyO; GMR-Gal4; UAS-Buffy/CyO; Ddc-Gal4* and *UAS-Buffy/CyO; GMR-Gal4* complex lines were used to overexpress α-*synuclein* or *Buffy* in neurons and the developing eye and were produced employing standard homologous recombination and marker selection methods as previously described (*M'Angale & Staveley, 2016b*, *2016c*). Gel electrophoresis was used to detect the presence of PCR products.

### Ageing assay

Several crosses of each genotype were performed and male flies collected upon eclosion and assessed using a protocol previously described (*M'Angale & Staveley, 2016a*; *Todd & Staveley, 2012*). For each genotype, at least 200 flies were aged and scored every 2 days for the presence of deceased adults (*Staveley, Phillips & Hilliker, 1990*). Survival data was analysed using GraphPad Prism version 5.04, and curves were compared using the Log-rank (Mantel–Cox) test with statistical significance determined at 95%, at a $P \leq 0.05$ with a Bonferroni correction.

### Climbing assay

A cohort of the critical class male flies was collected upon eclosion and scored for their ability to climb using a method that was previously described (*Todd & Staveley, 2004*). Climbing analysis was performed using the GraphPad Prism version 5.04 and climbing curves were fitted using non-linear regression and compared using 95% CI with a $P$ value of 0.05 or less being statistically significant.

### Scanning electron microscopy of the Drosophila eye

Male flies were collected upon eclosion and aged for up to five days and then prepared for scanning electron microscopy using a standard protocol as previously described (*M'Angale & Staveley, 2016a*). For each genotype, at least 10 different eye images were analysed using the National Institutes of Health ImageJ software (*Schneider, Rasband & Eliceiri, 2012*) and biometric analysis performed using GraphPad Prism version 5.04. Disruption area of the eye was calculated as has been previously described (*M'Angale & Staveley, 2012*). Statistical comparisons comprised one-way analysis of variance (ANOVA) and Dunnett's multiple comparison tests. $P$ values less than 0.05 were considered significant.

## RESULTS

### Drosophila *BI-1* is closely related to the human homologue

The 245 amino acids Drosophila BI-1 isoform A has a 42% identity and 68% similarity to the 295 amino acids human isoform B. The Drosophila homologue has a BI-1 domain between amino acids 21–223 and the human version at 74–286 (Fig. 1) as determined by the NCBI CDD (*Marchler-Bauer et al., 2015*). An alignment of the protein sequences using Clustal Omega (*Goujon et al., 2010*; *Sievers et al., 2011*) shows high conservation of the BI-1-like domain in the organisms analysed (Fig. 1A). Six TM domains in both Drosophila and human BI-1 were identified using both ELM (*Dinkel et al., 2014*) and TMpred (*Artimo et al., 2012*). An analysis of membrane-spanning domains by Phyre2 (*Kelley et al., 2015*) reveals seven TM domains (Fig. 1B) in both sequences that are highly identical in the cytoplasmic to intracellular orientation. An inhibitor of apoptosis binding motif at amino acids 1–5, an ER retention motif at position 221–224, and binding motifs for Atg8 at position 212–224 and calmodulin at amino acids 226–242 were identified by ELM. The presence of NES was detected in both Drosophila and human BI-1 using NetNES (*la Cour et al., 2004*) and only in Drosophila using the ELM. The 3D

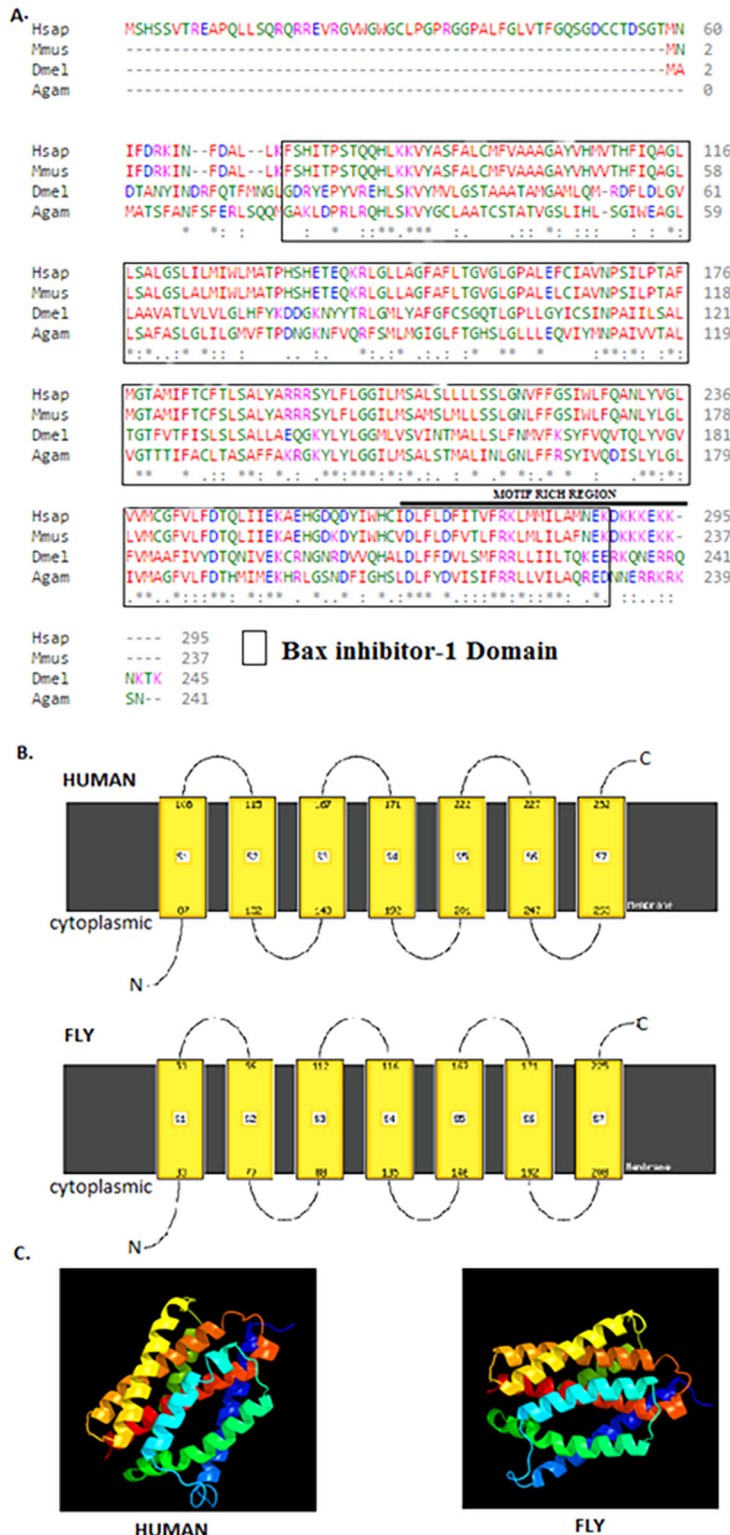

**Figure 1 Drosophila BI-1 has six TM domains that are evolutionarily conserved.** (A) The Drosophila, human, mouse, and mosquito homologues contain a BI-1 domain with the Drosophila version situated between amino acids 21–223 and the human version at 74–286 as determined by the NCBI Conserved Domain Database (*Marchler-Bauer et al., 2015*). They have six transmembrane-spanning regions as predicted by the Eukaryotic Linear Motif (ELM) (*Dinkel et al., 2014*) and TMpred (*Artimo et al., 2012*). It shows presence of a motif rich region, that contains a NES, ER retention motif, Atg8, and calmodulin binding motifs as identified using ELM. Sequence alignment was performed by Clustal Omega (*Goujon et al., 2010*; *Sievers et al., 2011*) and showed high conservation of the Bax inhibitor-1 domain (Hsap is *Homo sapiens* NP_001092046.1, Mmus is *Mus musculus* NP_001164506.1, Dmel is *Drosophila melanogaster* NP_648205.1, and Agam is *Anopheles gambiae* XP_315790.3). "*" indicate the residues that are identical, ":" indicate the conserved substitutions, "." indicate the semi-conserved substitutions. Colours show the chemical nature of amino acids. Red is small hydrophobic (including aromatic), Blue is acidic, Magenta is basic, and Green is basic with hydroxyl or amine groups. (B) Additional protein analysis performed using Phyre2 (*Kelley et al., 2015*) revealed the presence of seven transmembrane domains in both the Drosophila and human sequences (Image cartoons are obtained from Phyre2). (C) The 3D modelling of the Drosophila and human proteins using Phyre2 shows a close similarity in the structure and the orientation of the transmembrane domains with the image coloured by rainbow from the N → C terminus (Image cartoons are obtained from Phyre2).

modelling of these proteins using Phyre2 (Fig. 1C) shows a close similarity in the structure and the orientation of the TM domains with the image coloured by rainbow from the N → C terminus.

## Knockdown of *BI-1* in DA neurons decreases lifespan and severely impairs locomotor function

The expression of both *BI-1-RNAi* lines in the *Ddc-Gal4*-expressing neurons results in decreased lifespan and impaired locomotor function. The median lifespan for these flies was 54 days for *BI-1-RNAi (1)* and 46 days for *BI-1-RNAi (2)* when compared to 70 days for the controls that express the *lacZ* transgene as determined by the Log-rank (Mantel–Cox) test (Fig. 2A). When *BI-1* is suppressed in these neurons, the flies develop an early onset impairment of locomotor ability as determined by the nonlinear fitting of the climbing curves (Fig. 2B). The 95% CI for the slope were 0.033–0.050 and 0.0175–0.0355 for the two RNAi lines respectively when compared to 0.052–0.070 for the *lacZ* control flies. These results appear to suggest a role for *BI-1* in the protection of neurons in Drosophila.

### *Buffy* suppresses the *BI-1-RNAi*-induced phenotypes

The directed overexpression of the pro-survival Bcl-2 homologue *Buffy* results in increased lifespan and improved climbing ability (*M'Angale & Staveley, 2016a*). When *Buffy* is co-expressed with both *BI-1-RNAi* lines in the *Ddc-Gal4*-expressing neurons, the results indicate an increased median lifespan of 70 days and 72 days respectively when compared to 74 days for *Buffy* co-expressed with *lacZ* control flies and 70 days for the *lacZ* flies, as determined by Log-rank test (Fig. 3A). The climbing ability of the *BI-1-RNAi* flies was not significantly different from the *Buffy* co-expressed with *lacZ* controls as determined by comparison of the *BI-1-RNAi* climbing curves (Fig. 3B) with the control curve. The 95% CI for the slope of *BI-1-RNAi (1)* was 0.0340–0.057 and that of *BI-1-RNAi (2)* was 0.040–0.061 when compared to 0.035–0.050 and 0.052–0.070 for the controls. Taken together these results suggest a pro-survival role for *BI-1*; as the phenotypes induced by its knockdown are significantly counteracted by the pro-survival *Bcl-2* homologue *Buffy*.

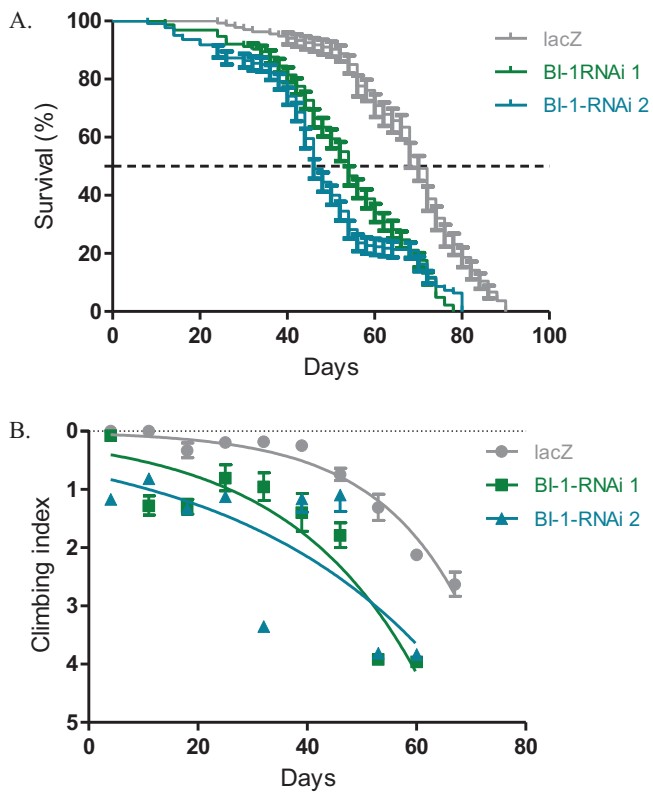

**Figure 2 Loss of *BI-1* activity decreases survival and impairs climbing ability.** (A) The inhibition of *BI-1* in the *Ddc-Gal4*-expressing neurons results in reduced lifespan when compared to control flies expressing the *lacZ* transgene. The genotypes are *Ddc-Gal4/UAS-lacZ*, *Ddc-Gal4/UAS-BI-1-RNAi 1*, and *Ddc-Gal4/UAS-BI-1-RNAi 2*. Longevity is shown as percent survival ($P < 0.05$, determined by the Log-rank (Mantel–Cox) test and $N \geq 200$). (B) The inhibition of *BI-1* in these neurons resulted in a significant decrease in climbing ability as determined by nonlinear fitting of the climbing curves and comparing the 95% CI. The genotypes are *Ddc-Gal4/UAS-lacZ*, *Ddc-Gal4/UAS-BI-1-RNAi 1*, and *Ddc-Gal4/UAS-BI-1-RNAi 2*. Error bars indicate standard error of the mean and $N = 50$.

## Knockdown of *BI-1* with the expression of α-*synuclein* slightly alters phenotypes

The expression of α-*synuclein* in dopaminergic neurons results in impaired locomotor function that is attributed to cellular toxicity. The co-expression of *BI-1-RNAi* along with α-*synuclein* in the *Ddc-Gal4*-expressing neurons, slightly exacerbated the reduced survival and the loss in climbing ability observed with the expression of α-*synuclein*. The median lifespan was 52 days and 54 days for flies that express *BI-1-RNAi* along with α-*synuclein* compared to 58 days for controls that co-express α-*synuclein* along with the *lacZ* transgene and 70 days for flies that express the benign *lacZ* transgene (Fig. 4A) as determined by Log-rank test with $P < 0.001$. A comparison of the climbing curves by nonlinear fitting at 95% CI revealed they were significantly different (Fig. 4B), with a CI of 0.038–0.049 for *BI-1-RNAi (1)* and 0.025–0.033 for *BI-1-RNAi (2)* co-expressed along with α-*synuclein* and compared to 0.052–0.069 for the α-*synuclein* co-expressed with *lacZ* control flies. This implies that the knockdown of *BI-1* in the *Ddc-Gal4*-expressing neurons abrogates its cytoprotective function and enhances the α-*synuclein*-induced phenotypes.

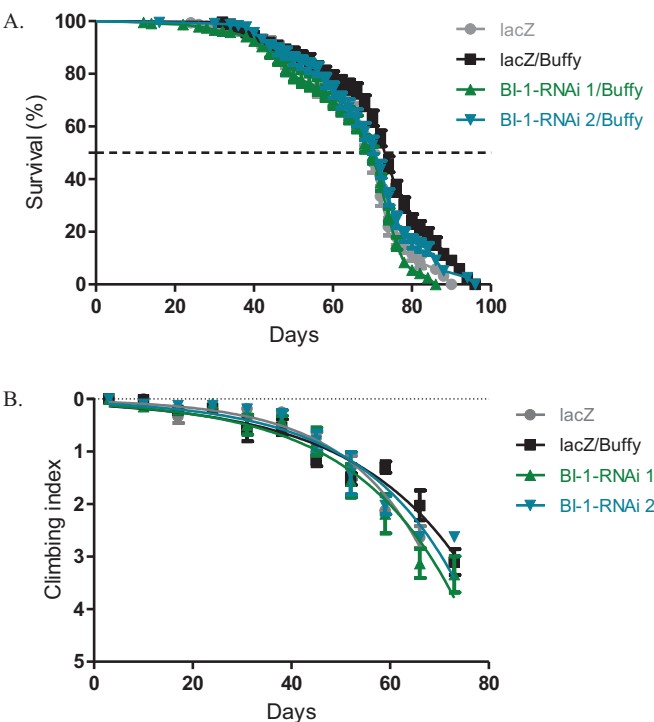

**Figure 3 The *BI-1*-induced phenotypes can be suppressed by the overexpression of *Buffy*.** (A) The co-expression of *Buffy* with *BI-1-RNAi* in the *Ddc-Gal4*-expressing neurons result in the inhibition of the observed phenotype of decreased survival when compared to the control. Genotypes are *UAS-Buffy; Ddc-Gal4/UAS-lacZ, UAS-Buffy; Ddc-Gal4/UAS-BI-1-RNAi 1*, and *UAS-Buffy; Ddc-Gal4/UAS-BI-1-RNAi 2*. Longevity is shown as percent survival ($P < 0.05$, determined by Log-rank (Mantel–Cox) test with $N \leq 200$). (B) The inhibition of *BI-1* along with the overexpression of *Buffy* in these neurons results in the suppression of the age-dependent loss in climbing ability. The genotypes are *UAS-Buffy; Ddc-Gal4/UAS-lacZ, UAS-Buffy; Ddc-Gal4/UAS-BI-1-RNAi 1*, and *UAS-Buffy; Ddc-Gal4/UAS-BI-1-RNAi 2*. Analysis was done by nonlinear fitting of the climbing curves and significance was determined by comparing the 95% CI. Error bars indicate standard error of the mean and $N = 50$.

## Knockdown of *BI-1* in the eye decreases ommatidia number and increases degeneration, phenotypes that are rescued upon *Buffy* overexpression

The directed knockdown of *BI-1* in the Drosophila developing eye using the *GMR-Gal4* transgene resulted in eyes with decreased number of ommatidia and a higher disruption of the ommatidial array in both the RNAi lines that were tested (Figs. 5A–5C and 5J) as determined by a one-way analysis of variance with a *P* value $< 0.0001$. Co-expression of both *BI-1-RNAi* lines with *Buffy* restored the mean number of ommatidia to control levels as determined by a one-way analysis of variance with $P = 0.2439$ and $0.2342$. The percentage disruption was significantly different from the control flies (Figs. 5D–5F and 5K). Taken together, these results suggest that BI-1 may play a pro-survival role in the development of the Drosophila eye and that *Buffy* suppresses the developmental eye defects that result from the knockdown of *BI-1*. The knockdown of *BI-1* along with α-*synuclein* expression resulted in a significant decrease in the number of ommatidia or increase in percentage disruption of the eye as determined by a one-way analysis of

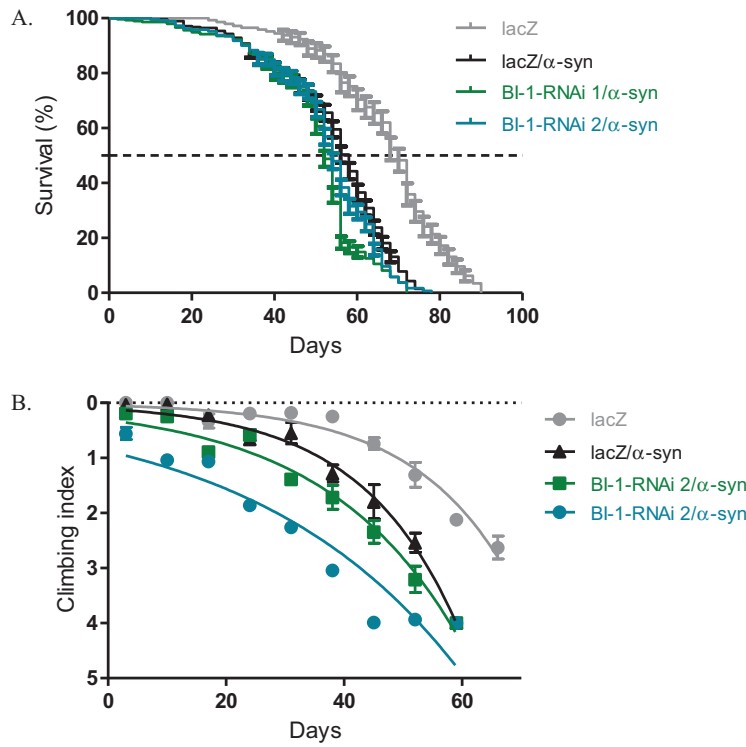

**Figure 4 Loss of *BI-1* in neurons complements the α-*synuclein*-induced phenotypes.** (A) The inhibition of *BI-1* along with α-*synuclein* expression in the *Ddc-Gal4*-expressing neurons resulted in a shortened lifespan when compared to the control. Genotypes are *UAS*-α-*synuclein*; *Ddc-Gal4/UAS-lacZ*, *UAS*-α-*synuclein*; *Ddc-Gal4/UAS-BI-1-RNAi 1*, and *UAS*-α-*synuclein*; *Ddc-Gal4/UAS-BI-1-RNAi 2*. Longevity is shown as % survival (*P* < 0.05, determined by Log-rank (Mantel–Cox) test with *N* ≤ 200). (B) The co-expression of *BI-1-RNAi* with α-*synuclein* resulted in a slight but significant decrease in the age-dependent loss in climbing ability when compared to the control. The genotypes are *UAS*-α-*synuclein*; *Ddc-Gal4/UAS-lacZ*, *UAS*-α-*synuclein*; *Ddc-Gal4/UAS-BI-1-RNAi 1*, and *UAS*-α-*synuclein*; *Ddc-Gal4/UAS-BI-1-RNAi 2*. Analysis was done by nonlinear fitting of the climbing curves and significance was determined by comparing the 95% CI. Error bars indicate standard error of the mean and *N* = 50.

variance with a *P* value < 0.0001 in both instances (Figs. 5G–5I and 5L). The number of ommatidia and percentage of disruption was worse than with either α-*synuclein* expression or *BI-1* knockdown. This indicates that the knockdown of *BI-1* enhances the α-*synuclein*-induced eye defects.

## DISCUSSION

The knockdown of *BI-1* via stable inducible RNAi in the *Ddc-Gal4*-expressing neurons of Drosophila results in decreased survival and impaired climbing ability over time. Although there is no known homologue of *Bax* in Drosophila, the only pro-apoptotic *Bcl-2* homologue is *Debcl* (*Brachmann et al., 2000*; *Colussi et al., 2000*; *Igaki et al., 2000*; *Zhang et al., 2000*), and has been demonstrated to possess pro-apoptotic functions. The Drosophila BI-1 is able to block *Bax*-induced cell death in yeast (*Chae et al., 2003*), and reduction of *BI-1* function induces cell death (*Xu & Reed, 1998*). These results suggest neuronal dysfunction may result from degeneration or death when the function of *BI-1* is reduced in the *Ddc-Gal4*-expressing neurons. The *BI-1*-induced cell death could occur

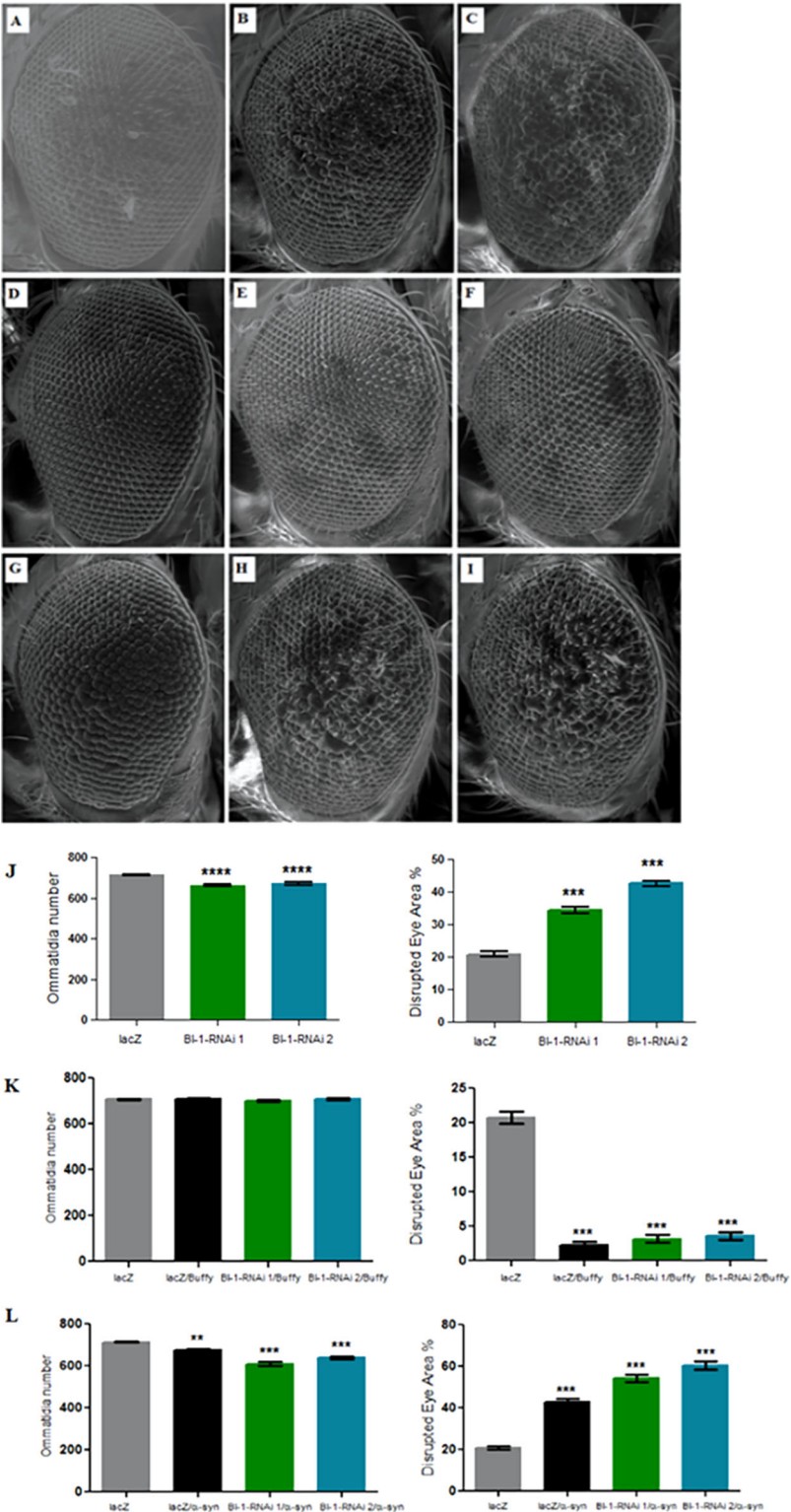
**Figure 5 Knockdown of *BI-1* in the developing eye results in decreased ommatidia and increased degeneration of the ommatidial array.** Scanning electron micrographs when *BI-1* is repressed in the Drosophila developing eye; (A) *GMR-GAL4/UAS-lacZ*, (B) *GMR-GAL4/UAS-BI-1-RNAi (1)*, and (C) *GMR-GAL4/UAS-BI-1-RNAi (2)*, when repressed along with overexpression of *Buffy*; (D) *UAS-Buffy; GMR-Gal4/UAS-lacZ*, (E) *UAS-Buffy; GMR-Gal4/UAS-BI-1-RNAi (1)*, and (E) *UAS-Buffy; GMR-Gal4/ UAS-BI-1-RNAi (2)* and when co-expressed with α-*synuclein*; (G) *UAS-α-synuclein; GMR-Gal4/UAS-lacZ*, (H) *UAS-α-synuclein; GMR-Gal4/UAS-BI-1-RNAi*, and (I) *UAS-α-synuclein; GMR-Gal4/UAS-BI-1-RNAi*. (J) Biometric analysis when *BI-1* is repressed in the eye indicated decreased ommatidia number and higher percentage of ommatidial disruption when compared to the control. (K) The co-expression of *Buffy* with both *BI-1-RNAi* lines resulted in the suppression of the eye phenotypes, the ommatidia number and disruption of the eye were restored to control levels. (L) The knockdown of *BI-1* along with α-*synuclein* expression resulted in worsened eye phenotypes, the number of ommatidia was lower and the degree of ommatidial disruption was higher than either the knockdown of both *BI-1* lines or that of α-*synuclein* when compared to controls. Comparisons were determined by one-way analysis of variance (ANOVA) with a Dunnett's multiple comparison post-test ($P < 0.05$), error bars are standard error of the mean, $N = 10$ and asterisks represent statistical significance (*$P < 0.05$, **$P < 0.01$, and ***$P < 0.001$).

through interaction with pro-survival Bcl-2 proteins at the ER membrane (*Xu & Reed, 1998*) and especially Bcl-2 and Bcl-X$_L$ in humans and possibly Buffy in Drosophila. BI-1 seems to be involved in cellular functions that are protective to ER stress-induced apoptosis (*Chae et al., 2004*). It seems to do this by the regulation of calcium ions (*Lisak et al., 2016*; *Xu et al., 2008*) and ROS (*Kim et al., 2009*; *Lee, Kim & Chae, 2012*). BI-1 regulates ER stress by controlling ER-generated ROS accumulation and stress linked to the UPR. Therefore, the knockdown of this important ER stress regulator in the DA neurons would result in neuronal degeneration and death. The only pro-survival *Bcl-2* homologue in Drosophila is *Buffy* (*Quinn et al., 2003*) and the overexpression of *Buffy* is known to confer survival advantages to cells under normal conditions and under conditions of stress (*Clavier et al., 2014*; *M'Angale & Staveley, 2016a*, *2016c*, *2017*; *Monserrate, Chen & Brachmann, 2012*; *Quinn et al., 2003*; *Sevrioukov et al., 2007*). The overexpression of *Buffy* along with the knockdown of *BI-1* resulted in the suppression of the *BI-1*-induced phenotypes. This Buffy action may be specific to its interaction with BI-1 or to its general pro-survival signalling pathways. The rescue of the *BI-1*-induced phenotypes in both the *Ddc-Gal4*-expressing neurons and in the developing eye may indicate a pro-survival role for *BI-1* in Drosophila, as the pro-survival action of *Buffy* can abrogate its phenotypes.

The expression of human α-*synuclein* in DA neurons of Drosophila results in impaired climbing ability (*Feany & Bender, 2000*), similar to what is observed in reduced *BI-1* function. The expression of α-*synuclein* along with the reduction of *BI-1* activity significantly altered the impaired locomotor ability observed. The age-dependent reduction of climbing ability could be a result of *BI-1*-induced apoptosis coupled with neurotoxicity that result from α-*synuclein* accumulation and the subsequent dysfunction of cellular mechanisms. Although we observed the enhancement of the α-*synuclein*-induced phenotypes by the knock down of *BI-1*, marked by the reduction in longevity and a precocious loss in climbing ability, the *BI-1*-induced phenotypes were hardly altered by the expression of α-*synuclein*. This observation may exclude the involvement of α-*synuclein* in the BI-1 pathway as its expression does not enhance the

BI-1 phenotypes, and the observed phenotypes may be because of the knockdown of *BI-1*. All the same, it appears that the presence of either of the mechanisms, vis-a-vis *BI-1*-induced apoptosis or *α-synuclein* aggregation neurotoxicity, confers a great disadvantage to *Ddc-Gal4*-expressing neurons.

The suppression of *BI-1* in the Drosophila eye under the direction of the *GMR-Gal4* transgene results in a lower ommatidia number when compared to the control. *BI-1* is an apoptosis suppressor gene and the down-regulation of its protein product results in programmed cell death (*Li et al., 2014*). The reduction in the ommatidia number observed is mainly due to the fusion of ommatidia and the resulting ommatidia disarray. The knockdown of *BI-1* in the Drosophila eye seems to exacerbate the *Gal4*-induced apoptosis that manifests as roughened-eye phenotype (*Kramer & Staveley, 2003*). The co-expression of the *Bcl-2* pro-cell survival homologue *Buffy* with *BI-1-RNAi* results in the suppression of the phenotype, with the number of ommatidia and the roughened eye restored to control levels. Buffy seems to ameliorate this phenotype and it is possibly via a general action on survival signals or an interaction with BI-1.

The expression of *α-synuclein* in the Drosophila eye results in reduced ommatidia and a highly disrupted ommatidial array (*Feany & Bender, 2000*). This *α-synuclein*-induced developmental eye defects model is a viable system to show the effects of altered gene expression and its role in neuroprotection. The co-expression of *α-synuclein* with *BI-1-RNAi* in the Drosophila eye resulted in decreased ommatidia number and a highly disrupted ommatidial array when compared to the control that expresses *α-synuclein*. The number of ommatidia decreased further when *α-synuclein* was co-expressed with *BI-1-RNAi*. Additionally, the degree of disruption of the ommatidial array was also increased. Though it did not appear to be additive in nature, it seems that the combination of the expression of the neurotoxic *α-synuclein* and the knockdown of the activity of the anti-apoptotic BI-1 results in a worsening of the roughened eye phenotype. The accumulation of *α-synuclein* has been implicated in breakdown of cellular homeostasis that include apoptosis, ROS production, and autophagy (*Chinta et al., 2010*). The knockdown of *BI-1* disrupts regulation of similar mechanisms as those implicated in *α-synuclein*-induced neurotoxicity that include apoptosis, autophagy, and ROS production (*Li et al., 2014*). It therefore, follows that the combined action of *α-synuclein* expression and *BI-1* knockdown worsened the phenotypes that result from either *α-synuclein* expression or *BI-1* knockdown.

## CONCLUSION

The knockdown of *BI-1* in the Ddc-Gal4-expressing neurons of Drosophila results in reduction in lifespan and an age-dependent loss in climbing ability, phenotypes that are strongly associated with the degeneration and loss of dopaminergic neurons. The co-expression of the pro-survival *Buffy* with *BI-1-RNAi* results in the rescue of the phenotypes observed, it is possible that Buffy and BI-1 participate in cellular pathways that promote anti-apoptosis. Finally, *BI-1* appears to be neuroprotective as its knockdown along with *α-synuclein* expression result in enhanced phenotypes.

### Funding

P. Githure M'Angale has been partially funded by Department of Biology Teaching Assistantships and a School of Graduate Studies Fellowship from Memorial University of Newfoundland. The research program of Brian E. Staveley has been funded by the Natural Sciences and Engineering Research Council of Canada (NSERC) Discovery Grant. The funders had no role in study design, data collection and analysis, decision to publish, or preparation of the manuscript.

### Grant Disclosures

The following grant information was disclosed by the authors:
Department of Biology Teaching Assistantships and a School of Graduate Studies Fellowship from Memorial University of Newfoundland. Natural Sciences and Engineering Research Council of Canada (NSERC).

### Competing Interests

The authors declare that they have no competing interests.

### Author Contributions

- P. Githure M'Angale conceived and designed the experiments, performed the experiments, analysed the data, contributed reagents/materials/analysis tools, wrote the paper, prepared figures and/or tables, reviewed drafts of the paper.
- Brian E. Staveley conceived and designed the experiments, contributed reagents/materials/analysis tools, reviewed drafts of the paper.

### Data Deposition

The raw data has been supplied as Supplemental Dataset Files.

### Supplemental Information

Supplemental information for this article can be found online at http://dx.doi.org/10.7717/peerj.2974#supplemental-information.

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
