# Peer review of "Bax-inhibitor-1 knockdown phenotypes are suppressed by Buffy and exacerbate degeneration in a Drosophila model of Parkinson disease"

_PeerJ, doi:10.7717/peerj.2974_

## Round 0.1 · original submission · Major Revisions

· Academic Editor

Major Revisions

All three reviewers have raised overlapping concerns about whether the controls for the knockdown and over-expression experiments are the most appropriate. This is a major point that should be addressed comprehensively in a resubmitted manuscript. Other concerns from Reviewers 1 and 2 about the way the article is pitched, including the accuracy of the title, should also be taken on board in a revised manuscript. The questions raised by Reviewer 2 about the fluctuations in variability in the climbing assay and the way the data are presented should also be addressed.

Reviewer 1 ·

Basic reporting

In the conclusion, the authors state ' it is possible that the Buffy and BI-1 protein interact to promote anti-apoptotic mechanisms'. Given the lack of biochemical data, the authors should consider re-wording to unambiguously differentiate protein-protein interaction from a possible interaction of the pathways.

Experimental design

1. Have the two RNAi lines been validated for efficient knock-down of BI-1 in other studies? If yes, can the authors cite those publications.
If not, the authors should validate efficient knock-down by either qRT-PCR or western blotting. If validation in vivo is technically difficult, the authors could use in vitro experiments (for example in HEK293T cells) to prove efficient knock-down. The consistent data from two different RNAi lines is suggestive of specific and sufficient knock-down, however a validation will help strengthen the data.

Validity of the findings

Figure 3. Buffy overexpression needs to be compared to a wildtype (lacZ line) to check for increases in lifespan or climbing ability by overexpressing Buffy. Similarly, the rescue of the RNAi phenotypes by overexpressing Buffy needs to be compared to wildtype flies (in this case the lacZ line) and RNAi alone flies to prove a rescue of phenotype.

Figure 4. Similarly, the phenotypes of the alpha-synuclein overexpression along with LacZ or RNAi 1/2 should be compared to wildtypes and RNAi alone expressing flies. It is important to compare the phenotypes of the RNAi only expressing flies to those coexpressing alpha-synuclein to analyze any additive effect of co-expressing alpha-synuclein to the knockdown of BI-1. From figure 1, the lifespan of RNAi1/2 flies was reported as 54 and 46 days respectively, while the lifespan of the RNAi1/2 flies co-expressing alpha-synuclein is reported as 52 and 54 days respectively. The authors should discuss why knocking down BI-1 enhances the alpha-synuclein phenotype while the over-expression of alpha-synuclein has no effect on the BI-1 knock-down phenotype.
The additive phenotype the authors discuss in the UAS-α-synuclein; Ddc-Gal4/ UAS-BI-1-RNAi 1 compared to the UAS-α-synuclein flies could be the phenotype of the Ddc-Gal4/ UAS-BI-1-RNAi 1/2 flies with no contribution from α-synuclein in this process. This would either argue againt a role for alpha-synuclein in the BI-1 pathway or place BI-1 downstream of alpha-synuclein in the pathway.
These data should be analyzed more in detail along with all the necessary controls and statistical analysis. The authors should discuss any possible inconsistencies or deviations from their original hypothesis. These controls should also be added for the locomotor assay. Should the authors choose to use their initial data from figure 2 as controls for figures 3,4, this should be clearly stated.
Figure 5. A similar comparison of all groups should be done to asses the effects of overexpressing Buffy alone or comparing RNAi phenotype to RNAi+ alpha-syn phenotype.

Additional comments

1. The link to the webpage describing the RNAi constructs "http://www.flyrnai.org/up-torr/GetSummaryByGene?organism=Fly " is incorrect.
2. The authors should cite the work showing that Buffy overexpression increases lifespan and improved climbing ability (line 217-218)

Reviewer 2 ·

Basic reporting

Throughout the article, the authors refer to phenotypes resulting from siRNA-mediated knockdown of BI-1 expression as resulting from “inhibition” of BI-1, or “loss of function” of BI-1. This terminology is imprecise and should be clarified to indicate that the phenotypes result from reduced BI-1 expression rather than inhibition or loss-of-function. The title, in particular, needs to be amended.

I am not convinced of the relevance of Figure1 to the research question being investigated and find some of the information presented in that figure misleading. The alignment shows conservation of BI-1 across Human, Mouse, Fly and Mosquito, which I find OK, but I am less enthusiastic about the motifs that have been annotated above the alignment. These are simply the output of online prediction algorithms, which I don’t feel should be published in figures with out experimental evidence confirming that they are relevant. Moreover, the predictions annotated above the alignments in FIG1A only describe 6 TM domains. However, the models presented in FIG1B and C, based on an alternate prediction algorithm, contain 7TM domains – which is more consistent with what is thought to be true for members of the TMBIM family. Given that the structure of BI-1 does not factor into the biological phenotypes investigated in the rest of the paper or their interpretation in the discussion, I feel that these data would be best left out. If the authors wish to highlight what is known about the structure of BI-1 and how that may relate to its function they would be better off referring in their introduction to the paper by Chang et al. (“Structural basis for a pH-sensitive calcium leak across membranes” Science, 2014), in which the structure of the bacterial homologue of BI-1 is solved and homology models for BI-1 based on this structure are presented.

Experimental design

My greatest concern with this manuscript is that the specific research question being investigated is not clearly articulated in either the abstract or the introduction. The introduction does state that the aim is to investigate the neuroprotective functions of BI-1 in Ddc-expressing neurons and in the developing eye of Drosophila melanogaster. However, it is not clear to me whether this addresses/fills a knowledge gap in the field. The influence of BI-1 on cell survival/cell death appears to have been widely studied in neuronal cells of other organisms and its activity has frequently been linked to Bcl2-regulated apoptosis. BI-1 has also previously been studied using Drosophila in other cell types/contexts as outlined by the authors in their introduction. I feel that the authors need to clarify how their study meets the second of the Experimental Design criteria and if so, improve the framing of their research question by revising the abstract/introduction.

Insufficient information is provided to interpret the results of the climbing assay. The protocol for this assay should be described in detail in the methods section.

Validity of the findings

Can the authors explain the variation in the climbing assay data in Fig2B? The data for this assay appears to follow reasonably smooth time-dependent trajectories in the other figures (3B and 4B), but for some reason jumps around a lot in this experiment? Why for example, would the values for d-BI-1-RNAi2 be very tightly clustered around 3.5 (ie. No visible error bars) at Day 32 and the recover to be tightly clustered around 1 at Days 39 and 46? The legend states N=50, but only 5 values are present for each sample/timepoint in the uploaded prism file. Can the authors clarify what the values in the prism file correspond to and how many flys were assayed at each timepoint?

It is not clear to me whether the most appropriate controls have been used for BI-1 knockdown experiments in which Flys expressing transgenic BI-1 RNAi are compared to a those expressing a lacZ transgene. Is there a reason that a fly line expressing a non-specific RNAi or one targeting an irrelevant gene has not been used?

Would it be possible to measure the extent to which BI-1 expression has been reduced in the transgenic flys? I appreciate that there are unlikely to be antibodies against drosophila BI-1, but if this could be done by qPCR it should be included.

The impact of BI-1 RNAi on the developing eye appears quite subtle. The data plotted to represent the impact on ommatidia number in Fig5J,K,L are shown on graphs whose y-axes start and finish at arbitrary numbers, which exaggerates the differences the authors wish to highlight. In my opinion, this is misleading and not best-practice. All of these plots should be amended so that the y-axis begins at zero and ends at the same value in all three cases. Likewise, uniform axes should be used for the Disrupted Eye Area (%) plots.

Reviewer 3 ·

Basic reporting

It is a very interesting study using fly as a model to study Parkinson's disease stress signaling, emphasizing the role of novel molecular targets in fly such as BI-1 and Buffy.

Experimental design

Although the experiment have been chosen and performed to detect several aspects of BI-1 and Buffy signaling in stress. I would like to stress on few points:
1. In Fig 2, although lacZ has been added as a control, a non-specific RNAi should be included in both experiments as additional control.
2. In Fig 3, lacz and Buffy alone should be added as a control to better understand the importance of Buffy signaling.
3. Similarly in Fig 4, synuclein and lacZ alone needs to incorporated as controls.

Basically controls need to incorporated to prove the significance of this study.

Validity of the findings

The above mentioned controls are required to figure out the actual validity of the findings. The statistical analysis has been carried out properly.

Additional comments

See above comments.

---

## Round 0.2 · Minor Revisions

· Academic Editor

Minor Revisions

The authors have produced a very good response to the reviewers' initial comments and their revised manuscript is much improved. To improve it further, I urge the authors to adopt the very minor suggestions put forward by Reviewer 2 and then the paper will be acceptable for publication.

Reviewer 1 ·

Basic reporting

No comments

Experimental design

No comments

Validity of the findings

No comments

Additional comments

I believe the authors have sufficiently answered all the concerns raised by the reviewers and the manuscript, in it's current form is suitable for publication.

Reviewer 2 ·

Basic reporting

no comment

Experimental design

no comment

Validity of the findings

no comment

Additional comments

I thank the authors for the effort they have made to incorporate reviewers comments and submit an improved revised manuscript for consideration.

I have two minor suggestions:

The statements added to the abstract and introduction have helped to clarify the aims of the study. However, the final sentence of the background section of the abstract stating that the authors “determined the effect of BI-1 knockdown on the a-synuclein-induced phenotypes” could be improved with additional context. At this stage of the article I do not think it will be clear to the uninformed reader what the a-synuclein induced phyotypes are or that a-synuclein overexpression constitutes a model of PD – though it is later borne out in the Results section. I feel the clarity would be improved if this were reorganized such that the authors provided context on the Drosophila PD model and why they wished to investigate BI-1 knockdown in this setting earlier in the abstract.

Also, as the pitch of the revised manuscript puts greater emphasis on the relevance of BI-1 knockdown in the context of their PD model, it might be appropriate to draw this out in the title. Perhaps I could suggest a further amendment to something along the lines of “Bax-inhibitor-1 knockdown phenotypes are suppressed by Buffy and exacerbate neurodegenration in a Drosphila model of Parkinson disease”.

I am happy with the rest of the changes made.

---

## Round 0.3 · accepted · Accept

· Academic Editor

Accept

I greatly appreciate the very constructive way you engaged with the reviewers to enhance the quality of your manuscript.